# A data science approach for multi-sensor marine observatory data monitoring cold water corals (*Paragorgia arborea*) in two campaigns

**Robin van Kevelaer**[1], **Daniel Langenkämper**[1], **Ingunn Nilssen**[2], **Pål Buhl-Mortensen**[3], **Tim W. Nattkemper**[1] *

1 Biodata Mining Group, Faculty of Technology, Bielefeld University, Bielefeld, Germany, 2 Research and Technology, Equinor, Trondheim, Norway, 3 Research Group Benthic Habitat, Institute of Marine Research, Bergen, Norway

* tim.nattkemper@uni-bielefeld.de

**Data Availability Statement:** All relevant numerical data are within the paper and its Supporting information files.

## Abstract

Fixed underwater observatories (FUO), equipped with digital cameras and other sensors, become more commonly used to record different kinds of time series data for marine habitat monitoring. With increasing numbers of campaigns, numbers of sensors and campaign time, the volume and heterogeneity of the data, ranging from simple temperature time series to series of HD images or video call for new data science approaches to analyze the data. While some works have been published on the analysis of data from one campaign, we address the problem of analyzing time series data from two consecutive monitoring campaigns (starting late 2017 and late 2018) in the same habitat. While the data from campaigns in two separate years provide an interesting basis for marine biology research, it also presents new data science challenges, like the the marine image analysis in data form more than one campaign. In this paper, we analyze the polyp activity of two *Paragorgia arborea* cold water coral (CWC) colonies using FUO data collected from November 2017 to June 2018 and from December 2018 to April 2019. We successfully apply convolutional neural networks (CNN) for the segmentation and classification of the coral and the polyp activities. The result polyp activity data alone showed interesting temporal patterns with differences and similarities between the two time periods. A one month "sleeping" period in spring with almost no activity was observed in both coral colonies, but with a shift of approximately one month. A time series prediction experiment allowed us to predict the polyp activity from the non-image sensor data using recurrent neural networks (RNN). The results pave a way to a new multi-sensor monitoring strategy for *Paragorgia arborea* behaviour.

## 1 Introduction

The cold water coral (CWC) *Paragorgia arborea* (common name "bubblegum coral", in the following referred to as *Paragorgia*) is a common species is the North Atlantic Ocean [1].

**Funding:** Authors TWN, RvK, DL received funding from Equinor ASA for carrying out the data analysis and editing the manuscript. The funders had no role in study design, data collection and analysis, decision to publish, or preparation of the manuscript.

**Competing interests:** The authors have declared that no competing interests exist.

*Paragorgia* forms colonies in the deep sea (below 200m) that are often fan-shaped. The colonies grow slowly, and it might take hundreds of years until a colony reaches its maximum size [2, 3]. A large colony (with a height of 4 m) has been reported to be 400 ± 100 years old [2]. As of 2021, *Paragorgia* is classified as "Near Threatened" in the Norwegian Red List for Species, based on a population reduction between 10% and 20% between 1960 and 2020 in Norway [4]. Human activities like fishing with bottom trawls threaten *Paragorgia* populations due to physical damage of colonies. Also, petroleum and aquaculture activities represent further potential threats to this coral species [4–7]. As *Paragorgia* provides habitat for various invertebrates and fish, either in the colonies themselves or between the colonies in colony aggregations (coral gardens) [6], a loss of *Paragorgia* colonies would have a negative impact on those species as well. To be able to distinguish between indirect impact, such as increased particle loads from drilling operations, normal variations in behaviour due to changes in parameters in the corals ambient environment is essential. Ocean current for example, is an important factor for food supply to *Paragorgia* and thus influences the location and shape of the colonies [2]. The results of a previous study on current-based organic matter transport at the LoVe observatory indicate that currents in this area make food particles from more distant sources available to the corals and also play an important role in the transport of organic matter down to the sea bottom [8]. Recent publications on *Paragorgia* and other CWCs indicate a strong relationship between current and coral activity as well, also in the short term [9, 10]. Further, polyp activity is found to be related to sensor data like chlorophyll, turbidity, depth (which can be used to obtain information on the tides), and temperature [11]. All these observations motivate the application of in situ marine imaging to record detailed digital photograph time series of coral colonies together with other data to reduce the gap of knowledge about *Paragorgia* in terms of behavior, spawning, and interactions with other species and to facilitate a sound species management. However, to investigate *Paragorgia* polyp behavior on this technical level, an efficient visual monitoring strategy and a specific data science approach are needed.

Modern sensor technology offers many possibilities to collect marine data. High resolution cameras can be used to record large amounts of image and video material while various other sensors collect information on chemical and physical oceanographic parameters [12, 13]. Besides attaching sensors to mobile platforms, e.g. autonomous underwater vehicles [14] or remotely operated vehicles [15], cameras and sensors can also be deployed in fixed underwater observatories (FUO) [16–19]. Especially FUOs cabled from shore for power supply and data transfer are well-suited for long-term *in situ* monitoring (i.e. weeks or months) of underwater ecosystems as they can provide data with high frequency and time coverage. Including high-resolution image and video data and acoustical, e.g. echo sounder data, these measurements can easily accumulate up to 10–50 GB of multimodal data per day.

The Lofoten-Vesteralen Ocean Observatory (LoVe) [18] is a FUO in the northeast Atlantic. Between November 2017 and April 2019, it has provided several thousand images showing a part of an ecosystem with *Paragorgia* colonies, recorded by the two optical sensors of a stereo camera mounted on a platform that allows for changing the viewing angle of the camera. In addition, a large amount of data was recorded by various environmental sensors (in the following referred to as non-image sensor data). Divided by a gap from July 2018 to December 2018, the high frequency data cover two time intervals of about eight and four months, respectively. Data from both the cameras and other sensors provide a valuable opportunity to compare trends and patterns in the ecosystem's data from two consecutive years but call for special approaches in its analysis due the heterogeneity in the data.

The term *data science* describes the scientific area dedicated to the development of interdisciplinary approaches integrating principles from computer science, statistics, and domain knowledge to understand and analyze real phenomena using large data collections [20, 21].

Multi-sensor data collections recorded with FUO (including images) are challenging as every sensor has its individual range, noise level, missing value problem, variance, temporal resolution etc. The digital HD images are a very special case as these must be processed by sophisticated algorithms in order to extract numerical data that represent valuable information about the objects of interest, like size measurements or changes in morphology or status. Here, computer vision methods, especially from the field of deep learning, offer high potential [22].

## 1.1 Computer vision for underwater observatory images

In one of the previous studies, Zuazo et al. [11] analyzed the polyp activity of a single *Paragorgia* colony at LoVe in an image time series recorded between February 2018 and June 2018. Their neural network classifies image pixels into one of three polyp activity categories or a background class with accuracies between 0.62 and 0.88 based on a small pixel neighborhood. Osterloff et al. [9] analyzed the activity of the reef-building CWC *Desmophyllum pertusum* (formerly *Lophelia pertusa*), applying a convolutional neural network (CNN). They used single-polyp annotations for their approach, focusing on a small part of a reef recorded by a fixed camera between 3 April 2015 and 10 November 2015. Also, longer term changes like tissue color change have been investigated using computer vision [9]. However, a broader biological interpretation of the results and observed trends is limited as the recorded data spans less than a year, which makes it impossible to assess inter-annual variations. Other ML applications to observatory image data considered either different species, like sponges [23] or much shorter time periods [10] and none of the works addressed the problem of analyzing data from two years, i.e. two campaigns. If data from two campaigns are to be analyzed, new questions to the computational methods must be raised. A machine learning-based computer vision must for instance be analyzed regarding its generalization performance. This addresses the question how well a deep learning network that was trained with data from one campaign generalizes to data recorded during the other campaign, i.e. its accuracy stays stable. If for instance the camera and / or light source position changes between two campaigns, the visual properties of the objects can change which would have an effect on the performance of the network. From a biological point of view, a good generalization performance is essential to avoid biases and data gaps, so reliable data describing physiology / morphology and / or taxonomy can be collected. Thus, it is essential as a prerequisite for a broad implementation and use of FUO. In case of the LoVe data recorded between 2017 and 2019 which is considered in this work, images were indeed recorded from different camera viewing angles and under varying illumination conditions. While repeated changes of the camera viewing angle enable the monitoring of a larger area, this creates a great challenge for computer vision as the polyps visual appearance is impacted. Thus, one main contribution of this paper is the development and evaluation of a new robust computer vision method to extract polyp activity values from images collected under these challenging conditions.

## 1.2 Polyp activity prediction using environmental sensor data

After extracting the polyp activity time series from the two campaigns and a correlation analysis with the sensor data, we address the question whether polyp activity can be predicted from the non-image sensor data using ML-based time series analysis or not. If for instance the camera is out of service due to errors or maintenance for some time, the gaps in image—derived polyp activity could be completed with these modeled values. Previously, two approaches for modeling *Paragorgia* polyp activity using non-image environmental sensor data as input have been published. Zuazo et al. [11] used a multilayer perceptron to predict a binary activity state from sensor data input. Like the images they used, the sensor data used by them represent a

portion of the sensor data we use in the experiments presented in this paper. Johanson et al. [10] modeled polyp activity as a 2-class problem as well, using logistic regression for classification. They used a comparably small dataset of 258 polyp activity values manually determined from images recorded between 5 June 2012 and 16 June 2012. Both approaches use single feature vectors, i.e. s set of measurements from one time point combined, as input. Johanson et al. also consider features with various time lags of several hours.

In contrast to previous works, we propose to use regression for predicting polyp activity, using non-image sensor data as input and the polyp activity at a given time point $a(t) \in [0: 1]$ (derived from the image from time point $t$) as a target output. As we assume that the activity state of a coral colony is affected by the development of the previous states of its environment rather than only the present state or a single previous state, we propose using Long short-term memory (LSTM) recurrent neural networks [24] for modeling the polyp activity. Furthermore, we evaluate the approach using data from two different deployments in 2017 and 2018.

## 2 Data collection

Images and environmental data were acquired in the Lofoten-Vesteralen Ocean Observatory (LoVe) Node 1, which is located 20 km from the coast of Norway, in a depth of about 258 m [18]. A cable for power supply and data transfer connects the FUO with a land station. At the time of data acquisition for this study, LoVe Node 1 consists of three platforms. The main platform is equipped with two acoustic Doppler current profilers (ADCP) for measuring current velocity and an echosounder. Furthermore, the observatory has two satellite platforms. *Satellite 1* carries a camera, a hydrophone, and sensors for measuring depth (pressure), temperature, chlorophyll, turbidity and conductivity. The *Satellite 2* contains a stereo camera with an adjustable viewing angle to record high-resolution digital images of *Paragorgias* and a camera flash. The list of sensors in the LoVe observatory can be found on the LoVe observatory website (https://loveocean.no/about-love). The viewing angle of the stereo camera can be changed, so that images from several perspectives can be recorded repeatedly. The two optical sensors of the stereo camera will be referred to as $K_0$ and $K_1$ in the following.

Both satellite platforms are located higher than the main platform. The distances between the main platform and Satellite 2 and between Satellites 1 and 2 are both about 100m. An overview and a map of this platform setup is given at the data portal website for LoVe observatory Node 1, through that all data used in this work are available [25].

In our experiments, we investigate two *Paragorgia* colonies shown by images $I_{i = 0, \ldots, 29812}$ recorded at Satellite 2 between November 2017 and June 2019 [26, 27], a red one in the foreground of the images and another colony in the background, which appears blue in the images. For sake of simplicity we will refer to these colonies sometimes as the "red coral" $C_r$ and the "blue coral" $C_b$. Both coral colonies are shown in Fig 1. Whether $C_b$ is present in an image depends on the camera position (see Fig 2). The object-camera distance of the red coral $C_r$ is lower than the object-camera distance of $C_b$, such that more details of $C_r$ are visible than of $C_b$. While the camera flash illuminates $C_r$ stronger in the upper parts of the colony, the illumination of $C_b$ is uniformly low.

The images have a size of $2206 \times 2752$ pixels. We preprocessed all images using gamma correction with $\gamma = 0.3$ as a nonlinear image color transformation in order to increase visibility, in particular for dark regions and the background of the images. For an example image with and without gamma correction, see S1 Fig.

From July 2018 to November 2018, no images are available. Based on that, we divide the data acquisition period into two intervals for separate analysis and comparison: Observation interval $\Gamma_1$ (from November 2017 to June 2018) and observation interval $\Gamma_2$ (from December

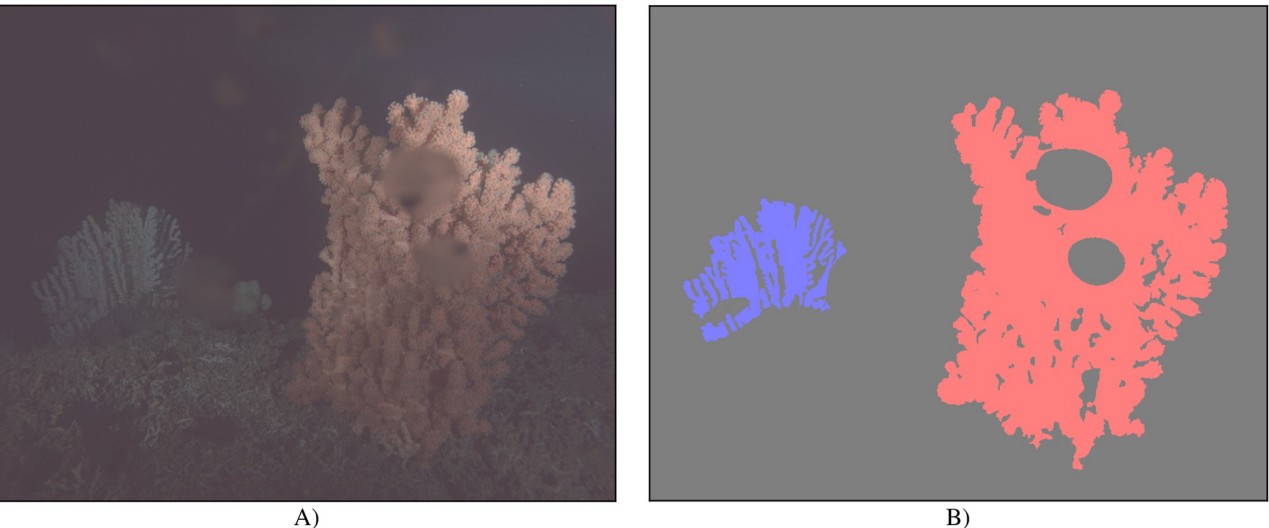

**Fig 1. Example image and ground truth segmentation mask.** A) Enhanced image (using gamma correction with $\gamma = 0.3$ for better visibility) and B) segmentation mask generated from BIIGLE annotations for A). In A), $C_b$ can be seen on the left, while $C_r$ can be seen on the right of the image in the foreground. In B), the annotated region for $C_r$ is shown in red, the region for $C_b$ is shown in blue, and the background is shown in gray. Biofouling in front of $C_r$ and the fish in front of $C_b$ are assigned to the background.

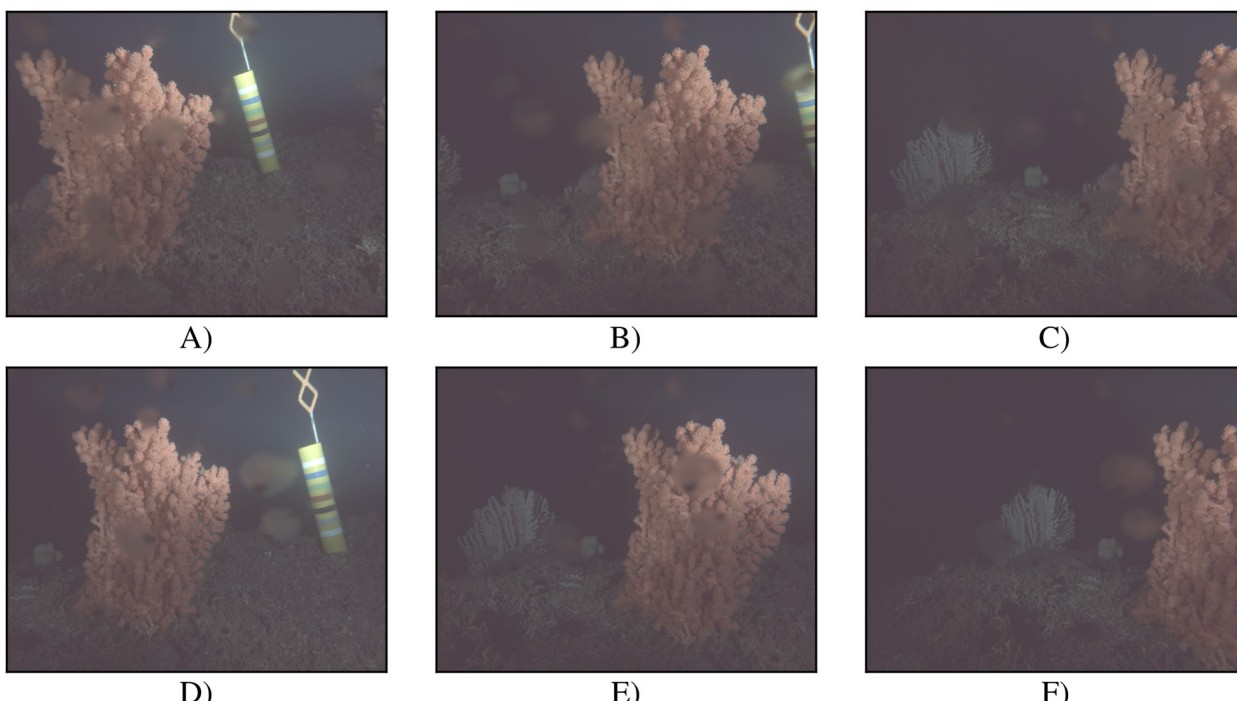

**Fig 2. Example images recorded at the LoVe observatory from different camera viewing angles.** Images A)—C) were recorded by stereo camera sensor $K_0$ and images D)—F) by sensor $K_1$, all during time period $\Gamma_2$. Images A) and D) are recorded from camera angle $\theta_0$, B) and E) from $\theta_1$, and C) and F) from $\theta_2$. In images A) and D), $C_b$ is not visible. In image B), only the rightmost part of $C_b$ is visible on the left edge. All images shown were enhanced using gamma correction with $\gamma = 0.3$ for better visibility.

2018 to June 2019). 15917 of the images were recorded during interval $\Gamma_1$ and 13896 images were recorded during interval $\Gamma_2$.

One image is available per stereo camera optical sensor per hour until 20 February 2018. After this date, the frequency is increased to two images, the camera switching between two viewing angles every two images. After 13 June 2018 (13:00), three images are available per hour for each stereo camera sensor, recorded from three different camera viewing angles that are repeated hourly. From 13 June 2018 on, information on the present camera orientation is available. The three different, hourly repeated camera angles used from that date on will be denoted as $\theta_0$, $\theta_1$, and $\theta_2$ in the following.

The later part of $\Gamma_1$, after 13 June 2018 (13:00), will be called $\Gamma'_1$ in the following. The camera setup used during $\Gamma'_1$ was reused throughout $\Gamma_2$. Example images recorded by stereo camera sensor $K_0$ during time period $\Gamma_2$ can be seen in Fig 2.

The two ADCP sensors at LoVe Node 1 measure the current velocity along three directions at different distances to the sea bottom [28]. We use the values measured nearest to the sea bottom by the Nortek Continental ADCP (190 kHz version) for our experiments [29–34]. The sensor starts measuring at a height of at least two meters above its position (blanking distance), facing towards the sea surface [28, 35].

Current data recorded before 7 May 2018 have a frequency of one set of measurements every ten minutes, i.e current measurements for the three directions and for each available distance to the sea bottom. In addition to the current data, we use temperature data and depth data [36, 37]. These data are available with an hourly resolution.

## 3 Methods

First, we will describe the pipeline of computer vision pipeline developed for estimating the average polyp activity in one *Paragorgia* colony at a time point $t$. The pipeline consists of three steps for coral colony segmentation, patch-wise polyp activity classification and average polyp activity estimation that are illustrated in Fig 3. Next, the result polyp activity time series will be analyzed for correlations and a polyp activity prediction will be proposed, based in non-image sensor data.

### 3.1 Data preparation

From the images recorded during observation interval $\Gamma_1$, we randomly select 60 images for training our models (dataset $\mathcal{I}_{1,\text{train}}$), another 20 images for validation ($\mathcal{I}_{1,\text{val}}$), and another 20 images for testing ($\mathcal{I}_{1,\text{test}}$). Analogously, we selected $\mathcal{I}_{2,\{\text{train,val,test}\}}$ from images recorded during

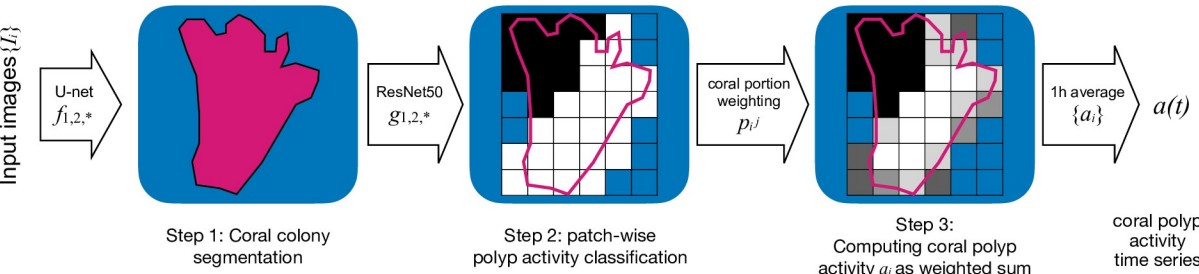

**Fig 3. Flow chart: Each image is processed in three steps to extract values describing the polyp activity in a coral.** First, A U-Net is used for coral segmentation. Second, image patches covering the coral region are classified as showing active or inactive polyps. Third, the patches' activities are weighted according the patches overlap with the coral segmentation. From all computed activity values $a_i$, a time series $a(t)$ with 1h resolution is computed.

interval $\Gamma_2$. In addition, we select a test dataset $\mathcal{I}_{1,\text{test}'}$ containing five images recorded during the special interval $\Gamma_1'$ to assess the effects of the setup change on 13 June 2018 (see Section 2) on the performance of our models. To train a baseline model with training data from both observation intervals, we define combined training and validation datasets $\mathcal{I}_{*,\text{train}} = \mathcal{I}_{1,\text{train}} \cup \mathcal{I}_{2,\text{train}}$ and $\mathcal{I}_{*,\text{val}} = \mathcal{I}_{1,\text{val}} \cup \mathcal{I}_{2,\text{val}}$. The shapes of the two *Paragorgia* colonies $C_r$ and $C_b$ are described in all training, test, and validation images with the BIIGLE [38] software using a free-hand polygon shape description tool. These descriptions are used as ground truth segmentation masks $M_i$ for images $I_i$ as a basis for segmentation learning and accuracy assessment. A mask image contains class labels at pixel positions as follows:

$$M_i(x, y) = \begin{cases} L(C_r) = 1 & \text{for colony } C_r \\ L(C_b) = 2 & \text{for colony } C_b \\ 0 & \text{else.} \end{cases} \tag{1}$$

Objects in front of the corals (e.g. fish, biofouling on the camera housing in front of the lens), larger, dark parts of the corals (e.g. branches shadowed by other branches) and background visible through the coral branches are also annotated as 0 and an example is shown in Fig 1.

## 3.2 Coral colony segmentation

Semantic segmentation with deep neural networks assigns a class label to each pixel in an image [39]. Here, we adapt the U-Net architecture [40] (implemented with [41]) for coral colony segmentation by implementing zero padding in the upsampling path. This enables the computation of an output mask $\hat{M}_i$ that matches the size of the input images, i.e each mask pixel $\hat{M}_i(x, y)$ corresponds to its counterpart in the corresponding input image $I_i$. One U-Net model is trained with each of the three training datasets $\mathcal{I}_{1,\text{train}}$, $\mathcal{I}_{2,\text{train}}$, and $\mathcal{I}_{*,\text{train}}$ to obtain three different segmentation models $f_1, f_2$, and $f_*$. We test each model using $\mathcal{I}_{1,\text{test}}$, $\mathcal{I}_{2,\text{test}}$, and $\mathcal{I}_{1,\text{test}'}$ to assess model generalization across time periods. For segmentation, all images are downscaled to a size of $551 \times 688$ to save memory and enable faster processing. Our training strategy also includes evaluating the model after each epoch using the validation dataset and keeping the best-performing model weights for testing and model application. Suitable hyperparameters are selected based on preliminary experiments (for details see S1 Text).

## 3.3 Patch-wise polyp activity classification

The polyp activity of an entire coral colony was computed as a weighted sum over the polyp activities computed for a set of image patches $D_j^i$, organized in a cartesian grid covering the segmentation mask $M_i$ of the coral in image $I_i$. Patch-wise polyp activity classification is explained for the observation interval $\Gamma_2$ data as this interval is not subdivided by different camera settings as $\Gamma_1$ (see Section 2). Again, individual models are trained for each coral and for all coral image data combined as a baseline.

All images $\mathcal{I}_{2,\{\text{train,val,test}\}}$ are subdivided into image patches $D_{j=0,\ldots,n_i}^i$ with a size of $128 \times 128$ pixels, where $n_i$ is the number of patches extracted from image $I_i$. All patches $D_j^i$ are assigned a coral portion $p_j^i \in [0, 1]$, which is defined as the overlap between patch $D_j^i$ and the segmentation mask $M_i$. For more details on patch generation and filtering see S2 Text. Patches $D_j^i$ with a coral portion $p_j^i < 0.1$ are not included in the training, test, or validation dataset.

To collect manual annotations for polyp activity, all patches in the training data are assigned class labels $k$ (with $k = 1$ for a patch showing active polyps and $k = 0$ otherwise) according to the following rule

$$k(D_j^i) = \begin{cases} 1, & \text{if } > 20\% \text{ of } C_r \text{ polyps shown in } D_j^i \text{ are extended } \wedge p_j^i \geq 0.1, \\ 0, & \text{else.} \end{cases} \quad (2)$$

In other words, if more than 10% of the patch pixels are covered with coral and at least 20% of the visible polyps are extended, this patch is labelled as a positive example. The threshold of 20% is set based on the visual experience obtained in the annotation in order to increase the confidence and reproducible class labels. An example showing coral surfaces with and without polyps as well as extended and retracted polyps can be found in Fig 4. Manual labeling was done by the author RvK who acquired knowledge about polyp activity classification in personal communication with the co-author PBM. Examples of annotated patches can be found in Fig 5. A patch $D_j^i$ with coral portion $p_j^i \geq 0.1$ is assigned to the classification training set $\mathcal{D}_{2,\text{train}}$, validation set $\mathcal{D}_{2,\text{val}}$, or test set $\mathcal{D}_{2,\text{test}}$ according to the assignment of its original image $I_i$ to $\mathcal{I}_{2,\{\text{train,val,test}\}}$. Datasets $\mathcal{D}_{1,\{\text{train,val,test,test}'\}}$ are collected analogously based on $\mathcal{I}_{1,\{\text{train,val,test,test}'\}}$.

A 50-layer deep residual neural network (ResNet-50) [42] is used for learning the patch classification task. ResNets are deep CNNs with residual blocks, that allow for building deep CNNs that can achieve high accuracy in various computer vision tasks. ResNets are commonly used, e.g. as a baseline in scientific studies, while improving them and their training methods is a subject of current research [43, 44]. We use the ResNet-50 code provided by PyTorch [41], initialized with weights pre-trained on the ImageNet dataset [45]. Instead of only adjusting a few final layers, we re-train all layers, which is called *fine-tuning* [46].

We train classification models, $g_\tau$, $\tau \in \{1, 2, *\}$, referring to data from different time intervals $\{\Gamma_1, \Gamma_2, * = \Gamma_1 \cup \Gamma_2\}$ using dataset $\mathcal{D}_{\tau,\text{train}}$. The $\Gamma$ in the index of $g_\tau$ is omitted for the sake of compactness. The models $g_\tau$ are applied to classify each patch $D_j^i$ with $p_j^i \geq 0.1$ as retracted polyps ($g_\tau(D_j^i) = 0$) or extended polyps ($g_\tau(D_j^i) = 1$) and evaluated using the test datasets $\mathcal{D}_{1,\text{test}}$, $\mathcal{D}_{2,\text{test}}$ and $\mathcal{D}_{1,\text{test}'}$.

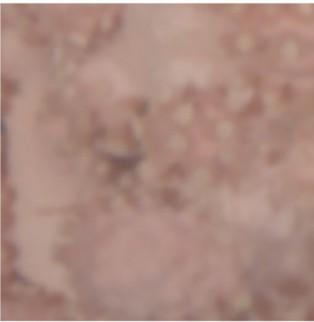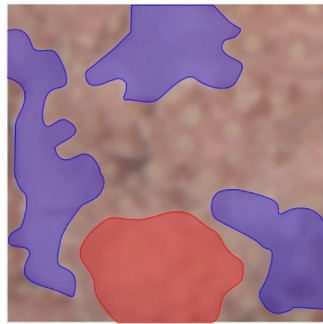

**Fig 4. Patch of $C_r$ with different polyp states.** Left image: Original patch. Right image: Polyp-bearing regions with retracted polyps marked red, non-polyp-bearing regions marked blue. In the unmarked regions, extended polyps can be seen (not fully extended).

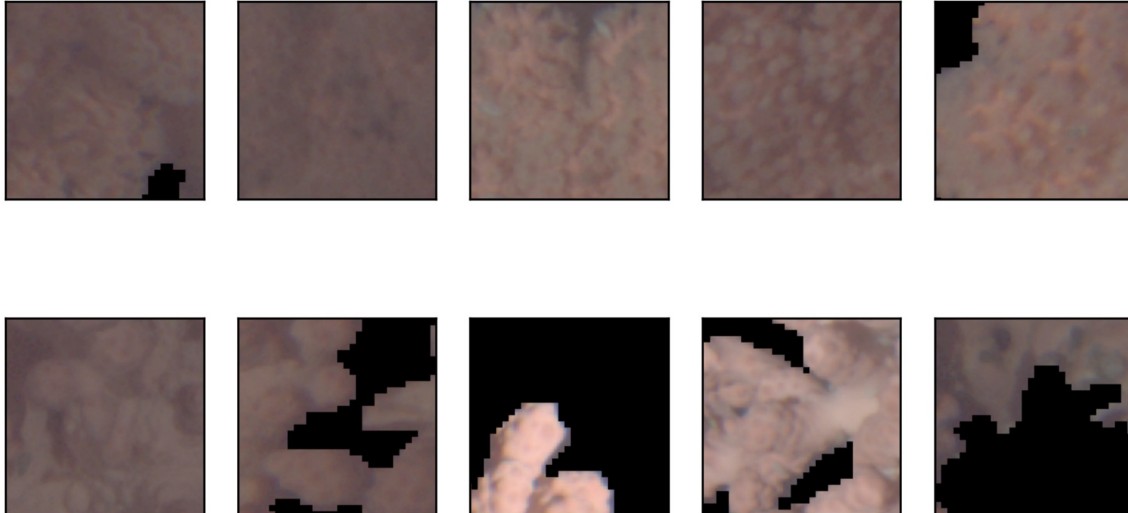

**Fig 5. Example patches of $C_r$.** The patches shown are extracted from various images in the datasets $\mathcal{D}_{1,\text{train}}$ and $\mathcal{D}_{2,\text{train}}$. The top row shows patches annotated as extended polyps (label 1), while patches in the bottom row are annotated as retracted polyps (label 0). The pixels annotated as background are set to (0,0,0), i.e. black pixels (see S2 Text).

## 3.4 Coral polyp activity computation

In the next step we compute coral polyp activity values $a(t)$ from the segmentation and patch classification results. A value $a(t)$ represents the average polyp activity in one colony at time point $t$, with a temporal resolution of one hour.

First, a segmentation model $f_\tau (\tau \in \{1, 2, {}^*\})$ is applied to the images to compute coral masks $\hat{M}_i$ in images $I_i$ first (see Section 3.2). The $\hat{}$ symbol is used to mark values, that have been computed by trained networks and are not directly derived from manual annotations. We generate image patches $\hat{D}_j^i$ showing parts of $C_r$ with coral portions $\hat{p}_j^i$, based on $I_i$ and $\hat{M}_i$. For further details on patch selection and coral portion computation, see S2 Text.

Next, a trained classification CNN model $g_\tau (\tau \in \{1, 2, {}^*\})$ is applied to assign each patch to a polyp activity class $\hat{k}(\hat{D}_j^i) \in \{0, 1\}$. For each image $I_i$, a polyp activity state $\hat{a}_i \in [0, 1]$ is computed as a weighted average of the patch labels $\hat{k}(\hat{D}_j^i)$ for one colony:

$$\hat{a}_i = \frac{1}{\sum_{j=0}^{n_i} \hat{p}_j^i [\hat{p}_j^i \geq 0.1]} \sum_{j=0}^{n_i} \hat{p}_j^i \hat{k}(\hat{D}_j^i) [\hat{p}_j^i \geq 0.1]. \tag{3}$$

By setting $\hat{p}_j^i = 0$ and $\hat{p}_j^i \hat{k}(\hat{D}_j^i) = 0$ if $\hat{p}_j^i < 0.1$, we ensure that patches $D_j^i$ with $\hat{p}_j^i < 0.1$ are not considered for computation of $\hat{a}_i$. In other words, for each patch with more than 10% coral pixels the polyp activity class (i.e. 0 (inactive) or 1 (active)) is determined. Those with a positive result contribute the overall activity sum, weighted according to their patch weight. the overall sum is normalized by the sum of all patch weights for a final weighted sum of coral colony activity.

For evaluation, the described segmentation and classification pipeline is applied to the test sets (results are given below). Ground truth data for the activity values per image $a_i$ are generated analogously, using manually generated segmentation masks $M_i$ and patch labels $k(D_j^i)$ as well as patches $D_j^i$ with $p_j^i$ generated based on masks $M_i$.

The final time series of activity values $a(t)$ has a temporal resolution of one hour. The $a(t)$ are computed as the average of the activity values $\hat{a}_i$ describing the polyp activity of $C_r$ in images $I_i$ in a one hour time window. Since the temporal resolution of imaging is not homogeneous during the campaigns this has to be done in a flexible way. For a given hour $t$, this includes all $\hat{a}_i$ for $I_i$ recorded at any time point $\tilde{t}$ with $t - 30$ minutes $\leq \tilde{t} < t + 30$ minutes (recorded by both optical sensors of the stereo camera). Only images recorded before 7 April 2019 are used for computing the activity time series as images are only sparsely available after that date.

To provide most accurate time series of polyp activities per time intervals $\Gamma_{(1,2)}$ and corals $C_{(r,\,b)}$, we use the models achieving the lowest MAE in this task (see Section 4.3). We generate the final time series as combinations of polyp activity data estimated by separate model combinations for the three time intervals $\Gamma_1$ $\Gamma_1'$, $\Gamma_1'$, and $\Gamma_2$. Input images for each hour $t$ were selected as described in Section 3.4.

**3.4.1 Polyp activity estimation for the blue coral $C_b$.**  Since the cameras are positioned to take good images of the "red" coral in the front, the blue coral $C_b$ is not visible in a subset of the images. As a consequence these images had be filtered out, which is described in detail in the supporting information (see S3 Text).

**3.4.2 Ground truth data and evaluation method for activity computation.**  To assess the accuracy of our approach we compare the activity values per image $\hat{a}_i$ computed with our approach with the manually obtained ground truth activity values $a_i$. As a performance measure, we use the mean absolute error (MAE). If a coral is present in an image for which no activity value is estimated (i.e. the coral was not found by segmentation model $f_\tau$), or if an estimated value is returned although the given coral in fact is not present in the image (i.e not annotated in the ground truth), the absolute error for this image is set to 1.

## 3.5 Evaluation metrics for segmentation and classification

The segmentation models $f$ are evaluated using the Jaccard score $J$ [47]. For evaluation of the classification models $g$, we use the $F_1$-score (also called F-measure), which is based on the precision ($P$) and recall ($R$) measures [48, 49].

To assess the overall performance of a segmentation or classification model, we use the macro-average of the performance measures introduced above. The macro-averaged performance measures $\bar{J}$ and $\bar{F}_1$ are unweighted average values of the performance measures of all classes [50–52]. A more detailed description of the performance scores and the macro average is given in S4 Text.

## 3.6 Polyp activity prediction using LSTM

To predict time series of hourly polyp activity values based on non-image sensor data, we use LSTMs (Long short-term memory). LSTMs are a special kind of recurrent neural networks and able to store past information in an internal state. Although first published in 1997 [24], LSTMs are still considered state of the art [53] in processing sequential data like sensor time series. We used the LSTM implementation from the PyTorch program library [41] to implement our polyp activity prediction models.

As a ground truth for LSTM training, we use time series $a(t)$ previously computed using our image-based approach described above. As input for the LSTMs, we define a sensor data time series $\mathbf{s}(t)$ that contains one vector of input features per hour $t$. Each feature vector $\mathbf{s}(t)$ is composed of the temperature $T(t)$, depth $d(t)$, and the current components $v_1(t)$, $v_2(t)$, and

$v_3(t)$:

$$\mathbf{s}(t) = (T(t), d(t), v_1(t), v_2(t), v_3(t)). \tag{4}$$

The current velocities are brought to an hourly resolution, assigning all values recorded at a time point $\tilde{t}$ with

$$t - 30 \ \text{minutes} \leq \tilde{t} < t + 30 \ \text{minutes} \tag{5}$$

to hour $t$. This is done in two steps: First, all values with a resolution of one second are brought to a resolution of 10 minutes, this way adjusting the resolution to the data recorded before 7 May 2018. Doing so, the time intervals

$$[t - 30 \ \text{minutes}, t + 30 \ \text{minutes}) \tag{6}$$

are subdivided into 10-minute intervals and the medians of the current data recorded within each of these intervals are computed. Second, one value per hour $t$ of the now fully 10-minute-resolved dataset is computed as the median of all data associated the one hour time window around $t$. For further details on the input features see Table 1. All input features are standardized to zero mean and unit variance.

To predict one activity value $\hat{a}(t)$, we use a sliding window approach, i.e. a subseries $\mathbf{S}_t = (\mathbf{s}(t - \eta + 1), \ldots, \mathbf{s}(t))$ of the sensor data series $\mathbf{s}(t)$ with a window length of $\eta$ hours as input for the LSTM. All values in $\mathbf{S}_t$ and their ordering can influence the output value $\hat{a}(t)$ due to the recurrence of the LSTM. LSTM models $h_{t'}$ are used to predict one week of hourly coral activity, where $t'$ is the first hour of the week to predict the data from. Each $h_{t'}$ is trained using data recorded during the eight weeks before $t'$. To predict the activity for the subsequent week, the window of eight weeks is shifted by one week and a new model is trained. This is repeated until all $\hat{a}(t)$, for which input series $\mathbf{S}_t$ are available, are computed. As the range of the output time series $\hat{a}(t)$ is known to be [0, 1], the LSTM outputs are clipped to this range at test time.

For the principles of feature and parameter selection of the LSTM, see S5 Text. For further details on data preprocessing, which includes the handling of gaps in the time series $\mathbf{s}(t)$, e.g. caused by sensor malfunctions, see S6 Text.

We apply polyp activity prediction using LSTM just for the front coral $C_r$ due to the large number of gaps in the $C_b$ polyp activity time series (for further details and the results for $C_b$ see S8 Text). We evaluate our LSTM approach using sets of polyp activity data $a(t)$ of coral $C_r$ as

**Table 1. Features used as input for LSTM-based activity prediction.**

| Data type | Variable | Resolution | Unit | Averaging |
|---|---|---|---|---|
| Temperature | $T(t)$ | 1 hour$^{-1}$ | °C | None |
| Depth | $d(t)$ | 1 hour$^{-1}$ | m | None |
| Current velocity, first component | $v_1(t)$ | $(10 \ \text{min})^{-1}$, 1 s$^{-1}$ | $\frac{m}{s}$ | Median |
| Current velocity, first component | $v_2(t)$ | $(10 \ \text{min})^{-1}$, 1 s$^{-1}$ | $\frac{m}{s}$ | Median |
| Current velocity, first component | $v_3(t)$ | $(10 \ \text{min})^{-1}$, 1 s$^{-1}$ | $\frac{m}{s}$ | Median |

Resolution refers to the resolution of the data downloaded from love.equinor.com. Averaging refers to the method used to change data resolution to 1 hour$^{-1}$. In case of current data, the medians are computed after bringing the data to a uniform resolution, using median computation as well.

ground truth, recorded during two time intervals:

$$\mathcal{A}_1 = \{a(\xi) \mid 12 \text{ April } 2018, \ 15:00 \ \leq \xi < \ 28 \text{ June } 2018\} \text{ and} \tag{6}$$

$$\mathcal{A}_2 = \{a(\xi) \mid 8 \text{ February } 2019, \ 19:00 \leq \xi < 7 \text{ April } 2019\}. \tag{7}$$

Activity values at earlier time points in $\Gamma_1$ or $\Gamma_2$ are excluded for model training as sensor data were not available. Performance evaluation of the presented LSTM approach is done using the MAE, which is 0.294 for $\mathcal{A}_1$ and 0.281 for $\mathcal{A}_2$.

### 3.7 Correlation analysis

Correlations are computed between the estimated polyp activity time series for corals $C_r$ and $C_b$ and between polyp activity and the non-image sensor data. We compute the Spearman rank correlation coefficient [54], which is applicable if the data can be ordered (ordinal variables). It is defined as follows [55]:

$$r_S(X, Y) = \frac{\sum_{i=1}^{N}(\text{rank}(X_i) - \mu(\text{rank}(X)))(\text{rank}(Y_i) - \mu(\text{rank}(Y))}{\sqrt{\sum_{i=1}^{N}(\text{rank}(X_i) - \mu(\text{rank}(X)))^2 \sum_{i=1}^{N}(\text{rank}(Y_i) - \mu(\text{rank}(Y)))^2}}, \tag{8}$$

where $X$ and $Y$ are vectors of length $N$ (in the following, these vectors will always be time series ordered by time), $\text{rank}(X_i)$ is the rank of $X_i \in X$ if $X$ would be sorted by value, and $\mu(\text{rank}(X))$ is the mean of all ranks of the elements in $X$. Only polyp activity and sensor data for time points (i.e. hours) for which a predicted polyp activity value is available are used for calculating correlations and, in case of current data, for standardization.

A two-sided t-test with the test statistic

$$t = r_S \sqrt{\frac{N-2}{1-r_S^2}} \tag{10}$$

($t$ is used here due to conventions for naming t-distributed test statistics, but is a different $t$ than is used as a variable for time in the rest of this paper) is used to compute $p$-values for the null hypothesis that X and Y are not correlated. The statistic is appropriate for larger samples ($> 30$) [55]. Spearman rank correlation and $p$-values were computed using SciPy (version 1.5.4) [56].

## 4 Results

### 4.1 Segmentation

The validation and test results of the segmentation models can be found in Table 2. An example segmentation result can be seen in Fig 6. The red *Paragorgia* $C_r$ was segmented with Jaccard scores greater than 0.88 in all test datasets. The blue *Paragorgia* $C_b$ was segmented with $J \geq 0.87$ if training and test dataset were from the same period ($\Gamma_1$ or $\Gamma_2$), except for test set $\mathcal{I}_{1,\text{test}'}$. Model $f_*$, trained using data from both time periods, achieved $J > 0.88$ for all datasets and both corals except for $C_b$ in $\mathcal{I}_{1,\text{test}}$ ($J = 0.828$).

The generalization performance of the U-Net was tested in cross-time interval segmentation experiments, i.e. $f_1$ was applied to the $\mathcal{I}_2$ and vice versa. In case of the "red" coral $C_r$ the generalization performance was very good, i.e. between 0.89 and 0.96. For the more distant $C_b$ the generalization was worse. In general, $f_2$ performend better on $\mathcal{I}_1$ than $f_1$ on $\mathcal{I}_2$.

The performance of $f_1$ segmenting $C_b$ in $\mathcal{I}_{2,\text{test}}$ is very low with a Jaccard score of 0.175

**Table 2. Jaccard scores per segmentation model and test dataset.**

| Model | Measure | Class $l$ | $\mathcal{I}_{1,\text{test}}$ | $\mathcal{I}_{1,\text{test}'}$ | $\mathcal{I}_{2,\text{test}}$ |
|---|---|---|---|---|---|
| $f_1$ | $\bar{J}$ | - | 0.947 | 0.840 | 0.672 |
| $f_2$ | $\bar{J}$ | - | 0.880 | 0.953 | 0.932 |
| $f_*$ | $\bar{J}$ | - | 0.925 | 0.961 | 0.939 |
| $f_1$ | $J_l$ | $L(C_r)$ | 0.962 | 0.951 | 0.886 |
| $f_2$ | $J_l$ | $L(C_r)$ | 0.949 | 0.956 | 0.941 |
| $f_*$ | $J_l$ | $L(C_r)$ | 0.964 | 0.960 | 0.943 |
| $f_1$ | $J_l$ | $L(C_b)$ | 0.896 | 0.612 | 0.175 |
| $f_2$ | $J_l$ | $L(C_b)$ | 0.718 | 0.922 | 0.870 |
| $f_*$ | $J_l$ | $L(C_b)$ | 0.828 | 0.939 | 0.888 |

For performance measures specific to a class label $l$, $l$ is given in the column "Class $l$" (see also Eq (1)).

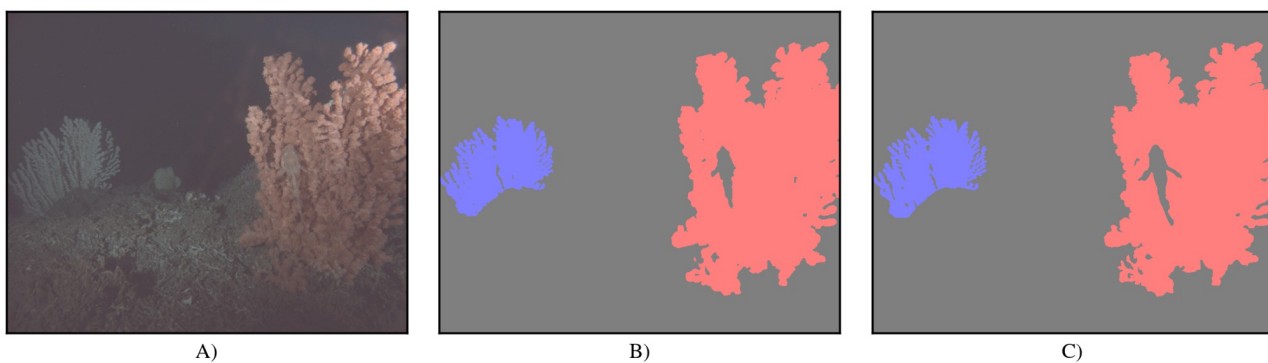

**Fig 6. Example result for image segmentation.** A): Original image. B): Segmentation result. C): ground truth mask. The image shown is taken from test set $\mathcal{I}_{1,\text{test}}$. Note that the fish in the foreground is not segmented as part of the coral, and also not annotated in the ground truth mask. The mask B) was generated using segmentation model $f_1$.

which is caused by special visibility conditions affecting the more distant $C_b$ that were not represented in the $\mathcal{I}_1$. Jaccard scores for the validation datasets as well as recall, precision and $F_1$ scores for the segmentation models can be found in S1 Table.

## 4.2 Patch-wise polyp activity classification

Test results of the classification models are listed in Table 3. ResNet50 classification models are denoted as $g_{\tau,c}$, where $\tau \in \{1, 2, *\}$ and $c \in \{r, b\}$ indicate time period $\Gamma_\tau$ and corals $C_{(r, b)}$,

**Table 3. Macro-averaged $F_1$ scores ($\bar{F}_1$) per test set, classification model, and coral.**

| Model | Coral | $\mathcal{D}^c_{1,\text{test}}$ | $\mathcal{D}^c_{1,\text{test}'}$ | $\mathcal{D}^c_{2,\text{test}}$ |
|---|---|---|---|---|
| $g_{1,r}$ | $C_r$ | 0.963 | 0.979 | 0.936 |
| $g_{1,b}$ | $C_b$ | 0.957 | 0.944 | 0.895 |
| $g_{2,r}$ | $C_r$ | 0.952 | 0.990 | 0.963 |
| $g_{2,b}$ | $C_b$ | 0.905 | 0.923 | 0.952 |
| $g_{*,r}$ | $C_r$ | 0.968 | 0.990 | 0.960 |
| $g_{*,b}$ | $C_b$ | 0.957 | 0.929 | 0.951 |

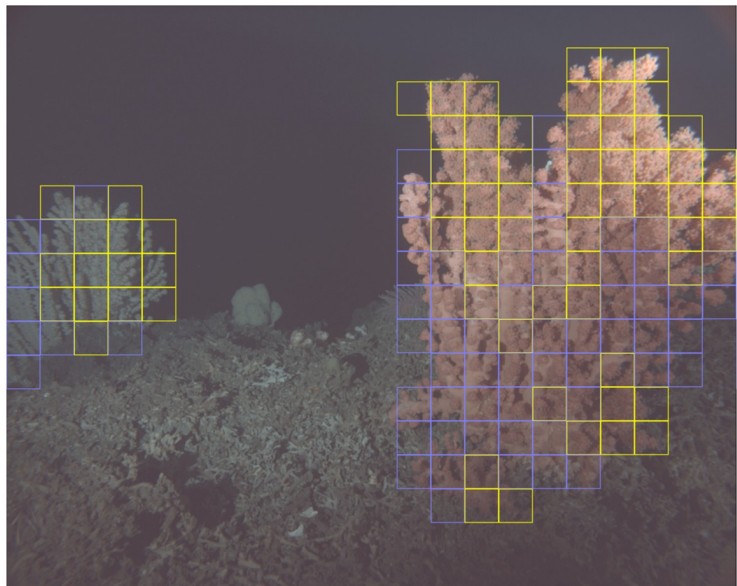

**Fig 7. Example patch classification result for $C_r$ and $C_b$.** Frames of extracted patches are shown in dark yellow if the patch was classified as showing extended polyps, while edges of patches classified as showing retracted polyps are colored blue. Gamma correction with $\gamma = 0.3$ was applied to improve visibility of the corals. Patches were generated based on masks segmented by model $f_1$. Patch classification was done using models $g_{1,r}$ and $g_{1,b}$ for $C_r$ and $C_b$, respectively.

respectively, $\Gamma$ and $C$ being omitted for a shorter notation. Classification of test data selected from the same time interval as the training data achieved high accuracy values $\bar{F}_1 > 0.94$ for all models $g_{\tau,\{r, b\}}$. Looking at the generalization performance across time intervals, we note the lesser performances for the blue coral ($\bar{F}_1$ of 0.895 and 0.905). For the "red" coral we observe higher cross interval generalization performances of 0.936 to 0.99. So called baseline models, trained on data from both time intervals, i.e $\Gamma_1$ and $\Gamma_2$, achieved $\bar{F}_1$ scores of at least 0.92 for all test datasets. An example showing patch classifications for both corals can be seen in Fig 7.

## 4.3 Coral polyp activity computation

The results of the evaluation of the coral activity estimation for the entire colonies can be found in Table 4. Again, for the red coral we observe very low error rates. The low generalization performance for the blue coral in the back suffers from the low quality segmentation result (see above). For the validation results, see S2 Table.

**Table 4. Mean Absolute Error (MAE) of computed coral polyp activity per image.**

| $f$ | $g$ | $\mathcal{I}_{1,\text{test}}$ | $\mathcal{I}'_{1,\text{test}}$ | $\mathcal{I}_{2,\text{test}}$ |
|---|---|---|---|---|
| $f_1$ | $g_{1,r}$ | 0.012 | 0.009 | 0.033 |
| $f_2$ | $g_{2,r}$ | 0.018 | 0.004 | 0.014 |
| $f_*$ | $g_{*,r}$ | 0.015 | 0.004 | 0.021 |
| $f_1$ | $g_{1,b}$ | 0.017 | 0.074 | 0.870 |
| $f_2$ | $g_{2,b}$ | 0.161 | 0.023 | 0.039 |
| $f_*$ | $g_{*,b}$ | 0.053 | 0.029 | 0.012 |

The table shows results for each test dataset and combination of segmentation and classification models ($f$ and $g$).

## 4.4 Computation time

The computation time for segmentation was approximately 45 images per second with a NVI-DIA Tesla V100 GPU since downscaled image versions were used (Downscaling speed was 1.45 images per second on an Intel Xeon processor). Polyp activity computation for the (larger) foreground red *Paragorgia* took 1.4 seconds per (original-sized) image, which includes generating patches and classifying them. For the blue *Paragorgia*, polyp activity computation speed was 3–10 images per second as the blue *Paragorgia* regions are smaller and only partially visible in some images, resulting in a smaller number of patches to be generated and processed.

## 4.5 Polyp activity time series analysis

The result time series can be seen for both corals and both time intervals in Fig 8. In addition to the hourly average activity values plotted in dots, we show a smoothed curve (Gaussian smoothing with $\sigma = 10$) visualizing a longer-term development of polyp activity over the months analyzed.

The plots show, that a large fraction of the computed polyp activity is either close to one or close to zero. This means that the polyps in one coral show a rather synchronized activity. Regarding the longer-term development of coral-specific polyp activity (smoothed curves), both corals show similar activity patterns. During $\Gamma_1$, the highest activity can be observed in

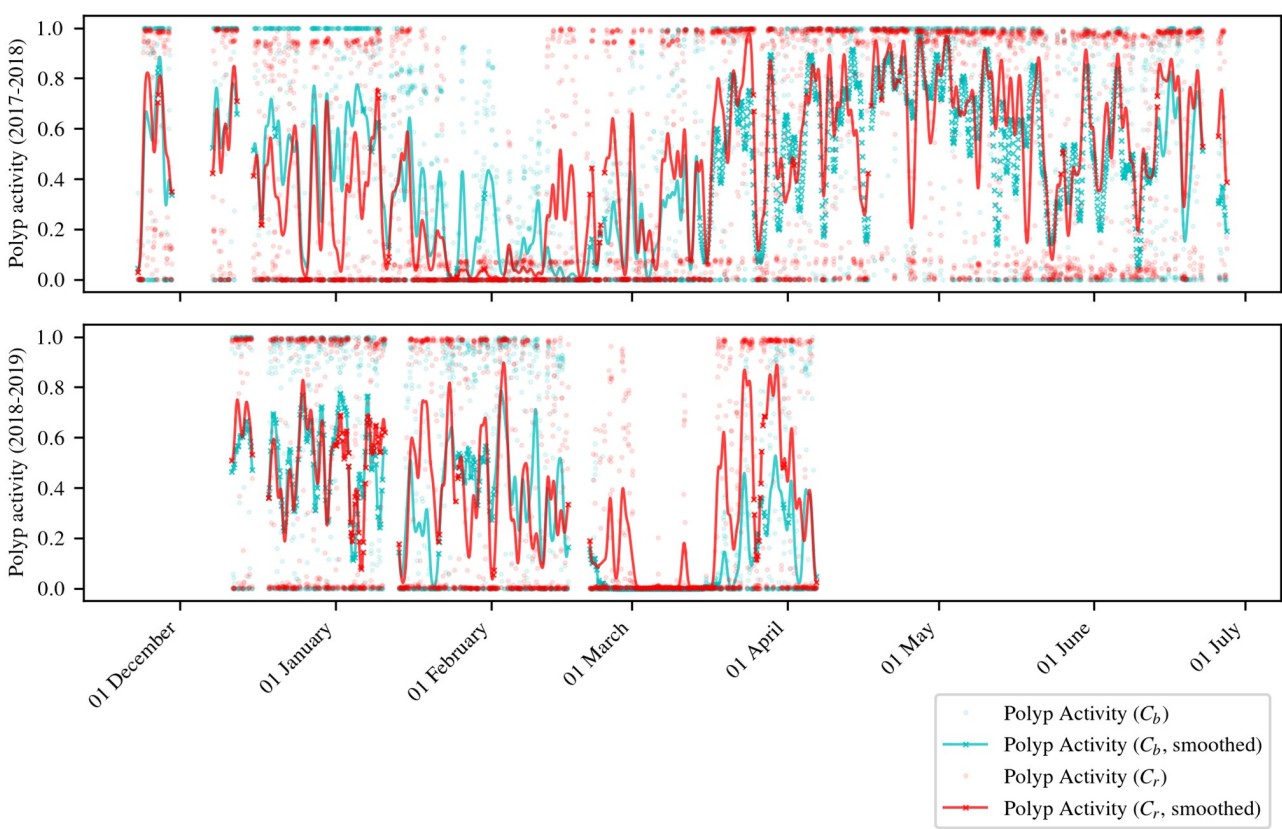

**Fig 8. Estimated coral activity vs. time.** For the smoothed curves, a Gaussian filter with $\sigma = 10$ was used. If no value is available for the next or previous hour, values in the smoothed curves are marked with an "x". For details on the handling of missing data before smoothing see S7 Text.

December and in May. After a short decrease in June, the activity increases again in July. During $\Gamma_2$, active and inactive periods alternate until late February. In March, the activity of the red coral $C_r$ is higher than the activity of the blue coral $C_b$.

In both time periods, and for both corals, periods of very low polyp activity occur (i.e. at most one single peak greater than 0.1 appears in the smoothed curve). For $C_r$, the low activity period in $\Gamma_1$ lasted from 22 January 2018 to 11 February 2018 (20 days). The low activity in $\Gamma_2$ lasted from 1 March 2019 to 17 March 2019 (16 days), beginning 38 days later compared to $\Gamma_1$. For $C_b$, the low activity period in $\Gamma_1$ lasted from 12 February 2018 to 20 February 2018 (8 days). The low activity period in $\Gamma_2$ lasted from 21 February 2019 to 19 March 2019 (26 days), beginning 9 days later compared to $\Gamma_1$.

The spearman rank correlation between the coral polyp activities of $C_r$ and $C_b$ is $r_S = 0.585$ ($p = 0.0$, $N = 3728$) for $\Gamma_1$ and $r_S = 0.525$ ($p = 2.2 \cdot 10^{-166}$, $N = 2347$) for $\Gamma_2$. For the calculation of correlations, only polyp activity values for hours $t$ for which a value is given for both corals (i.e. for each polyp activity value for $C_r$, a matching value for $C_b$ has to be given with respect to $t$, and vice versa). In S2 Fig we show plots of the polyp activity together with other sensor data (e.g. current, water depth).

**4.5.1 Influence factors for coral segmentation.** Although the full pipeline for coral polyp activity computation shows good performance values we investigated the effects of camera angle and biofouling in the U-Net segmentation. Fig 9 shows plots of the coral segmentation size (= number of pixels) for the blue coral $C_b$ during $\Gamma_2$ (the corresponding data for $C_r$ during $\Gamma_2$ is shown in S3 Fig). From the figures it can be taken that the visible region size depends on the camera angle. The region sizes in images in which the major part of the respective coral is visible decreased during time period $\Gamma_2$. Images showing both corals, recorded at different time points during $\Gamma_2$, and the respective segmentation masks are shown in Fig 10. In the shown images, biofouling on the camera housing can be seen, which increases over time. The

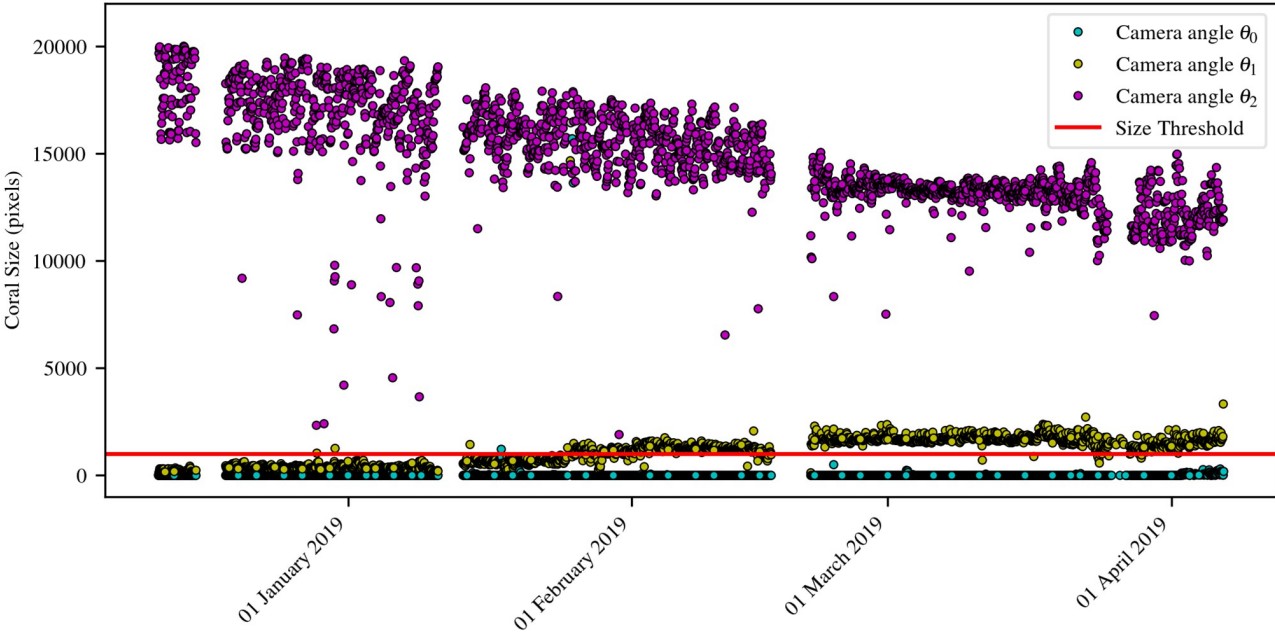

**Fig 9. Region size development of coral $C_b$.** The plot shows the region size development of $C_b$ during time period $\Gamma_2$ in the images recorded by stereo camera sensor $K_0$. Coral sizes in images recorded from different camera angles are marked by different colors. Region Segmentation was done using U-Net $f_*$. The lower threshold applied to remove false positive $C_b$ regions is shown as a red line.

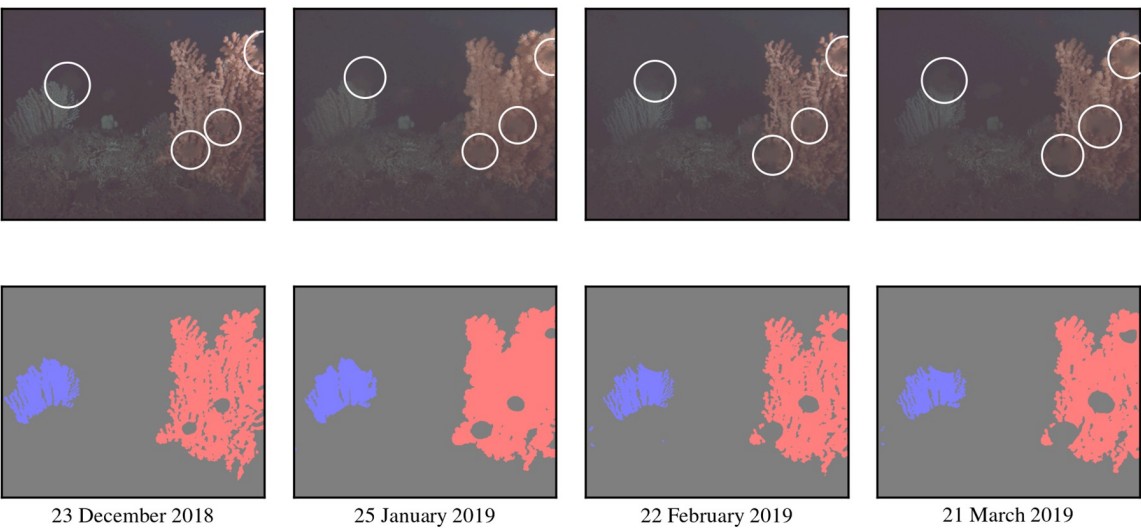

**Fig 10. Images and segmentation masks visualizing the development of biofouling.** The images are recorded at different time points during $\Gamma_2$ that are shown below each segmentation mask. The blurred spots in the images that are left out in the mask regions are biofouling spots. In the images in the upper row, biofouling spots overlapping with the corals are marked with white circles. The segmentation masks for each image were by generated by model $f_2$.

shown segmentation masks show that the models remove spots with biofouling from the segmented coral regions.

## 4.6 Polyp activity prediction using LSTM

The predicted polyp activity is compared to the polyp activity derived from the ground truth for $\mathcal{A}_1$ and $\mathcal{A}_2$. Plots of the curves are displayed in Fig 11). The plots show that the predicted time series basically follow the manually determined ground truth time series. For $\mathcal{A}_1$, it can be seen that the predicted curve sometimes stays on a high or low level instead of following short-term variations in the ground truth time series. For $\mathcal{A}_2$, the models reproduced the activity peak in the low activity interval in March 2019, however, two pairs of activity peaks were wrongly predicted during this interval. Furthermore, the predicted amplitude of activity peaks is often too low. Comparing the plots, the predicted and ground truth curves are more similar for $\mathcal{A}_1$. Plots of unsmoothed results and ground truth data can be found in S4 Fig.

Spearman correlations of the polyp activity time series intervals the datasets $\mathcal{A}_{(1,2)}$, are composed of with the predicted polyp activity and the sensor data time series used as LSTM input can be found in Table 5. Except for $v_3$ with $\mathcal{A}_2$, all p-values for current or temperature features are lower than $10^{-11}$. For these features, the absolute values of the correlations are between 0.2 and 0.45. The computed correlation values for depth are near zero, while the correlation value for $v_3$ and $\mathcal{A}_2$ is 0.107.

## 5 Discussion

In the first step of our approach we presented a new polyp activity estimation method for *Paragorgia* using deep learning-based segmentation, image patch classification and a special patch weighting scheme. We evaluated the method's performance on data recorded at the LoVe ocean observatory throughout two different deployments (i.e. time intervals) with different specifications and camera viewing angles. Two coral colonies were processed, one (referred to

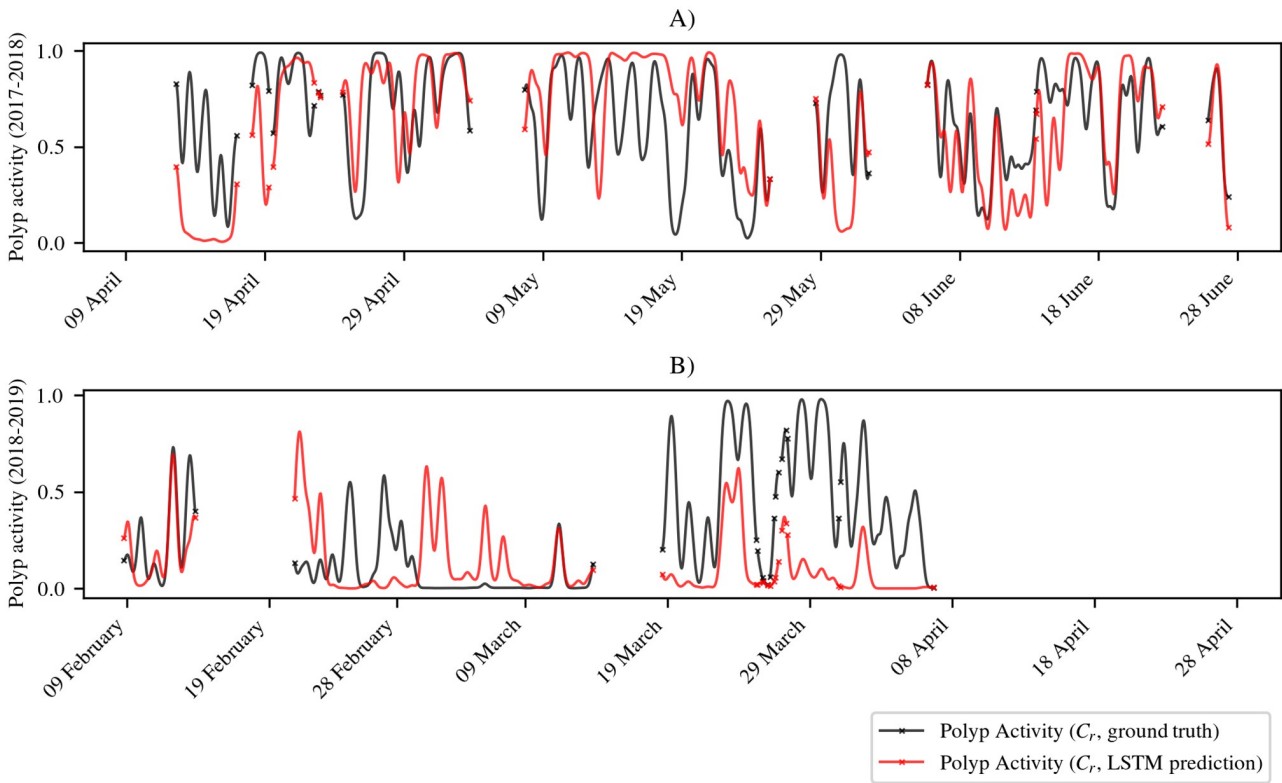

**Fig 11. Result of polyp activity prediction for the red *Paragorgia* $C_r$ using our LSTM approach.** Plots A) and B) show the ground truth and the results for $\mathcal{A}_1$ and $\mathcal{A}_2$, respectively. For better visibility, the original results are smoothed using a Gaussian filter with $\sigma = 5$. If no value is available for the next or previous hour, values in the smoothed curve are marked with an "x". For details on the handling of missing data before smoothing see S7 Text.

with $C_r$) shown in the foreground with good contrast and details and another one (referred to with $C_b$) in the background, shown with lower quality and details.

Our results for the red *Paragorgia* show that the method is able to estimate the polyp activity with high accuracy, even when the system is trained with data from the first deployment (November 2017—June 2018) and applied to the second deployment data (December 2018—April 2019), and vice versa (Jaccard score greater than 0.88 in segmentation and $F_1$ score greater than 0.93 in classification, see Tables 2 and 3).

**Table 5. Spearman rank correlations and p-values.**

| Datatype | $\mathcal{A}_1$ ($N = 1392$) | | $\mathcal{A}_2$ ($N = 1070$) | |
|---|---|---|---|---|
| | $r_S$ | $p$ | $r_S$ | $p$ |
| Depth | -0.049 | 0.066 | 0.020 | 0.505 |
| Temperature | -0.212 | $1.5 \cdot 10^{-15}$ | -0.338 | $4.5 \cdot 10^{-30}$ |
| Current velocity $v_1$ | -0.265 | $7.6 \cdot 10^{-24}$ | -0.203 | $2.1 \cdot 10^{-11}$ |
| Current velocity $v_2$ | 0.291 | $1.5 \cdot 10^{-28}$ | 0.438 | $2.8 \cdot 10^{-51}$ |
| Current velocity $v_3$ | -0.212 | $1.5 \cdot 10^{-15}$ | 0.107 | 0.0005 |
| Predicted activity | 0.460 | $8.4 \cdot 10^{-74}$ | 0.148 | $1.2 \cdot 10^{-6}$ |

The table shows the correlations $r_S$ between the features used as input for the polyp activity prediction LSTM and the ground truth polyp activity of coral $C_r$ in $\mathcal{A}_1$ and $\mathcal{A}_2$. In addition, the correlation between predicted and the ground truth-derived polyp activity is also contained in the table. All correlations are accompanied by p-values.

The segmentation accuracy for the blue *Paragorgia* is only slightly lower than for the red *Paragorgia* if training and test sets were recorded during the same deployment. An exception to this is the special case of a changed camera setup (see Table 2). Furthermore, in segmentation, the generalization performance between the two deployments was much lower for the blue coral than for the red one. An explanation for this is the higher object-camera distance and lower feature contrast due to lower illumination preventing the models from generalizing well, in particular when camera orientation changes. Classification accuracy for the blue *Paragorgia* is only slightly lower than for the red one, and the classification models also generalize well between deployments ($F_1$ scores greater than 0.89).

The segmentation results show that the size of the segmented regions for both corals decreased during the second deployment period (Fig 9, S3 Fig). Since we do not see a change in the shape of the corals in the second period, this observation is more likely to be explained by the FUO setup. Slight shifts of the camera occurred during deployment time, e.g. due to settling of the satellite platform on the seafloor and/or currents. The changed alignment of camera and flash can not only change the perspective but also reduce image brightness. Increasing biofouling on the camera housing in front of the lens reduced image quality over time, creating noisy spots in the field of view that occluded parts of both corals. The segmentation models were trained to exclude biofouling spots (see Section 3.1) from the segmented regions, which causes a decrease in size (i.e. number of pixels) of the segmented regions when biofouling increases (see Fig 10). These experiences should be considered in future setups and FUO designs if more than one colony is to be monitored. Of course, eliminating biofouling completely during longterm deployment is hardly possible, but it is essential to take precautions to avoid this as far as possible and to be aware of this problem when interpreting results.

And problems like biofouling can be avoided.

Regarding the aspect of efficiency we can report that our approach is able to process most images in less than 2.5 seconds. Thus, it is faster than the approach of [11], who reported a processing time of 75 seconds per image.

The polyp activities for both *Paragorgia* colonies are computed as time series $a(t)$, which approximate the degree of activity in a coral colony, based on many patches covering the *Paragorgia* segmentation. Using this approach, also a local polyp activity assessment (for instance at the top or at the bottom of the coral) could be performed. In the majority of images, most of the polyps within each of the two *Paragorgia* colonies had the same activity state (see time series in Fig 8). However, one strength of our approach is that it could also be applied for monitoring corals with polyps that are not synchronized, and show locally individual behavior i.e. feeding activity. In this case, more care should be given to the illumination to capture as much of the corals as possible in a way that the polyps can be recognized well.

The results of this image analysis were visualized as time series plots (see Fig 8). A striking feature of the time series generated for both observation periods is an interval of very low polyp activity during spring. This behavior appeared in both corals. Periods of inactivity lasted between 8 and 26 days and occurred later in the second observation period (approximately 5.5 weeks for the red *Paragorgia* and approximately 1.5 weeks for the blue *Paragorgia*, see Section 4.5). The underlying biological mechanisms of this behavior need to be further investigated. Also for the rest of the year, the activity of both coral colonies appears to be synchronized, which is indicated by the Spearman rank correlation computed for the two polyp activity time series (see Section 4.5).

Both time series appear synchronized although different classification models were used and the two corals have different visual properties. This indicates that the presented approach is capable of extracting information from lower quality data as well, here represented by the blue colony with a higher object-camera distance.

Polyp activity of the red *Paragorgia* from the time series intervals used for the LSTM regression experiments is slightly correlated with temperature and generally also shows a slight correlation with current velocity. The results of the LSTM time series prediction look promising and also indicate the relationship between current and polyp activity. However, the LSTM application was limited by many gaps in the non-visual sensor data.

General knowledge of how change in environmental parameters are impacting the natural behaviour of *Paragorgia* is a prerequisite to be able to monitor possible effects of human activities. As polypp activity is essential for feeding, it is regarded as an important parameter for measuring possible impact on health and used in laboratory experiments for establishing threshold values in risk assessment (include reference). As *Paragorgia* is hard to keep in aquarium, experience from field on what parameters that impact the organism is important to provide an optimal artificial environment.

## 6 Conclusion

Fixed underwater observatories (FUO) are applied for monitoring underwater habitats in various contexts. FUO equipped with cameras often record large amounts of image (and/or video) data that require classification and quantification of the information in the data before statistical analysis can be applied or observations can be related to other sensor data (such as temperature, current). We have demonstrated that our models can measure polyp activity from various deployments, which represents an important progress compared to the work of Osterloff et al. [9] and Zuazo et al. [11] as it shows that analyzing long-term monitoring data is possible without training new models for each observation period.

This progress paves the way for the application of deep learning-based image analysis to images collected with a group of FUO at different positions and/or in different time intervals to study one coral species (like *Paragorgia* in this case) or substructures of a colony without the need to collect manually annotated training data for each colony. It also indicates that transferring this approach to other CWC species has good chances of success. Monitoring at such a broad scale makes it easier to verify findings about observed events, which is often not possible if data from only one short observation period or location are available.

The statistical results correspond to observations published before hinting at a relationship between polyp activity and currents (e.g [9, 10]). In future work, the relationship of polyp activity time series to other parameters like the coral morphology or other environmental factors may be analyzed.

Using the polyp activity time series, we were able to train a LSTM to predict the polyp activity at a time point $t$ without integrating the image information. One potential application of such a prediction model could be filling in gaps in image-based activity time series, which is important as even with well-maintained equipment, long-time deployments always experience periods with camera failures, maintenance interruptions or biofouling. Such a prediction method could pave the way for new opportunities and approaches for interpreting environmental monitoring data.

## Supporting information

**S1 Fig. Example image with (A)) and without (B)) gamma correction with $\gamma = 0.3$.** The background of image B) is very dark, such that the corals, in particular $C_b$, cannot be visually assessed well.
(PDF)

**S2 Fig. Plots of polyp activity and sensor data.** The polyp activity time series $a(t)$ is shown together with the non-image sensor data used as LSTM input features (see Section 3.6).
(PDF)

**S3 Fig. Region size development of the red *Paragorgia*.** Size of the segmented $C_r$ region in the images recorded by stereo camera sensor $K_0$ during time period $\Gamma_2$ plotted against time. Regions were segmented by U-Net $f_2$. The sizes were plotted separately according to the camera angle of the corresponding images.
(PNG)

**S4 Fig. Unsmoothed LSTM result plot for the red *Paragorgia*.** Plot of unsmoothed ground truth and LSTM-predicted $C_r$ activity time series.
(PNG)

**S1 Text. Hyperparameter selection for segmentation and classification models.** A description of the parameter selection scheme used for the segmentation and classification models.
(PDF)

**S2 Text. Patch generation for polyp activity classification.** A description of the method used for generating image patches to be used as classification model input.
(PDF)

**S3 Text. Polyp activity estimation for the blue coral $C_b$.** A more detailed description of how the blue coral was processed.
(PDF)

**S4 Text. Detailed description of scores and performance measures.** Jaccard scores, precision, recall, and $F_1$ scores definition.
(PDF)

**S5 Text. Hyperparameter and feature selection for LSTM.** A description of parameter and feature selection scheme for the LSTM models.
(PDF)

**S6 Text. Preprocessing of sensor data in LSTM training and prediction.** Additional details of sensor data preprocessing, including a description how gaps in the sensor data time series are handled during LSTM training and prediction.
(PDF)

**S7 Text. Data smoothing for visualization.** A description of the smoothing process applied to polyp activity time series before visualization.
(PDF)

**S8 Text. LSTM prediction results and correlations for the blue *Paragorgia*.** LSTM prediction and evaluation results and correlations of LSTM input sensor data and polyp activity for the blue *Paragorgia* colony $C_b$.
(PDF)

**S1 Table. Additional evaluation results for the segmentation models.** Jaccard scores, precision, recall, and $F_1$ scores for test and validation of the segmentation models.
(PDF)

**S2 Table. Test and validation macro-averaged $F_1$ scores ($\bar{F}_1$) of the classification models.**
(PDF)

**S1 Data. Non-image sensor data and polyp activity time series.** Time series of non-image sensor data with an hourly resolution used in the experiments described in this paper and the result time series of polyp activity estimation.
(XZ)

**S2 Data. Annotations for the segmentation experiments.** Pixel-wise annotations used to generate ground truth segmentation masks for segmentation model training and evaluation.
(XZ)

**S3 Data. Ground truth data for the classification experiments.** Class labels for patches, weights for patches computed from ground truth masks, imagewise ground truth polyp states, patch extraction regions, and region sizes.
(XZ)

**S4 Data. Data for figures.** Non-image data that were used to generate figures shown in the paper, if not already contained in S1–S3 Data.
(XZ)

## Acknowledgments

We especially thank Geir Pedersen from the Institute of Marine Research, Bergen, Norway for providing valuable information about the ADCP current sensor and the current data format used at the LoVe observatory. The sensor data and images were downloaded from Equinor / IMR Ocean Observatory Archive, http://love.statoil.com/ from the sources and time periods listed in the Materials section and referenced in the Bibliography.

## Author Contributions

**Conceptualization:** Ingunn Nilssen, Pål Buhl-Mortensen, Tim W. Nattkemper.

**Formal analysis:** Daniel Langenkämper, Pål Buhl-Mortensen.

**Investigation:** Daniel Langenkämper, Ingunn Nilssen, Pål Buhl-Mortensen.

**Methodology:** Robin van Kevelaer, Daniel Langenkämper, Ingunn Nilssen, Pål Buhl-Mortensen, Tim W. Nattkemper.

**Project administration:** Ingunn Nilssen, Tim W. Nattkemper.

**Resources:** Ingunn Nilssen.

**Software:** Robin van Kevelaer, Daniel Langenkämper.

**Supervision:** Ingunn Nilssen, Pål Buhl-Mortensen, Tim W. Nattkemper.

**Validation:** Robin van Kevelaer.

**Visualization:** Robin van Kevelaer, Tim W. Nattkemper.

**Writing – original draft:** Robin van Kevelaer, Tim W. Nattkemper.

**Writing – review & editing:** Robin van Kevelaer, Daniel Langenkämper, Ingunn Nilssen, Pål Buhl-Mortensen, Tim W. Nattkemper.

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
