## [Decision Letter · Decision Letter 0]

25 Oct 2022

PONE-D-22-23889Computational analysis of multi-sensor marine observatory data monitoring cold water corals (Paragorgia arborea) in two campaignsPLOS ONE

Dear Dr. Nattkemper,

Thank you for submitting your manuscript to PLOS ONE. After careful consideration, we feel that it has merit but does not fully meet PLOS ONE’s publication criteria as it currently stands. Therefore, we invite you to submit a revised version of the manuscript that addresses the points raised during the review process.

We look forward to receiving your revised manuscript.

Kind regards,

Andrew Davies

Academic Editor

PLOS ONE

Journal Requirements:

Additional Editor Comments:

Thank you for your submission, this is interesting work and I have been able to secure two excellent reviewers who have provided constructive comments. I concur with both, please ensure that you improve accessibility of your manuscript and address the critical comment over the training of the model from the limited sample size.

Reviewers' comments:

Reviewer's Responses to Questions

**Comments to the Author**

1. Is the manuscript technically sound, and do the data support the conclusions?

Reviewer #1: Yes

Reviewer #2: Yes

2. Has the statistical analysis been performed appropriately and rigorously? 

Reviewer #1: Yes

Reviewer #2: Yes

3. Have the authors made all data underlying the findings in their manuscript fully available?

Reviewer #1: Yes

Reviewer #2: Yes

4. Is the manuscript presented in an intelligible fashion and written in standard English?

Reviewer #1: No

Reviewer #2: Yes

5. Review Comments to the Author

Reviewer #1: This is a detailed and interesting analysis that tackles the tricky problem of model generalization and accuracy under changing observation conditions (a common issue with long-term underwater image sets). I applaud the authors on their detailed explanation of the methods and application of models trained on one time period to data collected in another in order to demonstrate the general applicability of this approach.

My main issue with this study is that the model was trained over multiple time periods and conditions, but only on two individual organisms (one red and one blue Paragorgia), one of which it did not perform as well on for a number of reasons related to the quality and quantity of the data. This makes it somewhat harder to argue that this is a "generalized" model. The authors should be upfront about the fact that the model was trained on many varied observations of only two individuals, and the manuscript might benefit from framing this as a case study or proof-of-concept (especially given that the apparent correlation between polyp activity and conditions like temperature and current highlighted in the Discussion is based only on observations from one individual.)

I do not think that the limited number of individuals invalidates the approach or the value of this specific study, but it is an important caveat that deserves more acknowledgement early in the manuscript, ideally in the Introduction around lines 75 or 83 or at the beginning of the Materials/Methods (to the authors' credit, it is made clear in the abstract).

Below are observations for each section of the manuscript, with reference to specific line numbers as relevant.

Abstract:

"While some works of been published…" should be "have been published".

In general, I think the abstract could be made much shorter. There are sections that are redundant or too conversational for a concise summary of the study at hand.

Introduction:

The introduction suggests that the value of this type of study/methodology is to obtain more information on the ecology and behavior of Paragorgia via monitoring, but it is unclear how the data from a FUO is likely to contribute to this. It would be good to mention that this study is aimed at analyzing polyp activity specifically (and why we care about that) early in the introduction (closer to line 17 than line 83, where the subsection on this begins). The introduction fails to clearly establish a link between the need for better monitoring of Paragorgia and similar cold water corals and the way the data analyzed in this paper actually contributes to that. If the goal is to observe specific individuals and their feeding behavior (or other metrics like growth/survival/spawning/etc.) over time from a fixed location, this should be made clear (as well as how the proposed image analysis facilitates this).

The two subsections beginning at lines 50 and 83 are a confusing mix of details on prior models and approaches and more general information about the data being collected and the behavior and ecology of the target species. Many of the modeling details in the second subsection feel like they belong in the first, which suggests that maybe the biological subsection should be presented before the machine learning/monitoring subsection with details on analyzing the data. In general, I believe the introduction would benefit from reorganization so that 1) the biological relevance of collecting/analyzing this data is established earlier, and 2) the ecology of the target species is described separately from the technical details on past and present modeling efforts.

Line 4: Can you offer a more specific depth range for this species (or, barring that, a definition of the depths referred to as "deep sea")?

Lines 43-45 It is unclear why having multi-sensor data (e.g., temperature time series and HD images) precludes direct processing. I am not quite sure what the authors are trying to say here. There are a number of reasons why multiple data types might be challenging to process or combine, as well as other challenges to processing the sheer volume of observations obtained over a long deployment, but that is not quite what this sentence seems to be saying. This is an important statement to get right, since it establishes the necessity and value of the rest of this study's analysis/methodology. (See my comment on Lines 508-511.)

Materials:

Line 116: If the specific title "Materials" is not required for this section, a more descriptive title like "Data" or "Data Collection" could be more appropriate.

Methods:

Pay attention to tense throughout this section. Different subsections are written in different verb tenses, which is somewhat confusing, given that they all appear to describe different steps of the same analysis.

Lines 186 and 220: Given the number of images in the overall dataset, these seem like small numbers of images for training the model (although the level of detailed annotation required for each image may justify this). By my count, 205 total images were hand-classified for training, validation, and testing between the two time interval groups and the additional special interval Γ'1. Is this correct? If yes, it could be helpful to reiterate these total image counts on lines 218-222 when discussing manual labeling.

Results:

There are a number of subsections here where methods are reported alongside results. Methods should always be detailed in the Methods section and only restated here if necessary for understanding the results being reported. If these methods are being reported here because they apply to post-hoc analyses based on earlier results and were not part of the original study design, that should be made clear so that the reader can follow the rational for these additional analyses. I detail a couple of these instances below.

Lines 350-359: This section belongs in the Methods, not the Results. It is unclear to me why this entire subsection (beginning line 342) is in the Results and not the Methods, apart from the results discussed on lines 361-364.

Lines 366-370, 407-411 and 427-429: Again, these details should already have been covered in the Methods, although briefly restating them here to aid the reader in understanding the reported values may be okay.

Discussion:

Lines 432-436: While restating other details of the analysis, this would be a good place to include that observations were made of two individual colonies.

Line 451: These limitations that lead to lower accuracy for the blue Paragorgia are important to emphasize. Since only two individuals were analyzed here, it is likely that similar issues to those encountered with the blue Paragorgia would be relatively common should this model be applied to a larger dataset of different individuals in different conditions. While discussing how generalized the model is (regarding different time points and conditions), it is important to also acknowledge the limitations of this effort (regarding distinct individuals and poorer performance with distance, lighting, certain angles, etc.).

Lines 465-472: These details should ideally be reported in the Results, rather than the Discussion.

Conclusion:

Lines 508-511: I think this may be what lines 43-45 were trying to say.

Lines 524: While the results here are consistent with these previous observations, I do not believe they can be said to "confirm" such hypothesized relationships, given that the data comes from only one individual.

General Observations:

There is some inconsistency between the references to figure numbers in the text (e.g., Figure 1 vs. Fig. 1).

In general, this manuscript would benefit from proof-reading/copy editing to correct multiple small typos and cases of incorrect grammar. I have listed some lines on which I noted issues here, although the list is likely not comprehensive:

Lines 36, 38, 43, 71, 81, 85, 135, 179, 182, 183, 189, 218, 233, 241, 257, 290, 339, 477, 528, 532

Reviewer #2: This article answers a fundamental issue faced by scientists using images to monitor ecological phenomena: the bottleneck formed by data extraction from images. It presents a sophisticated method to automatically measure the level of soft corals polyps activity and then tests a modelling method to predict this activity as a function of the other parameters measured continuously by the observatory they are using. The manuscript is well written and draws appropriate conclusions. I see no issue with the method or the results and their interpretation. I, therefore, think it is worthy of publication.

However, I found this article hard to read given the complexity of the method and I found myself frequently coming back to the method section while reading the results to check the meaning of the many symbols, equations and formulas. This does not, by any mean, compromises the validity of the study and I acknowledge it is the way computer-science methods are reported, but I think additional comments in the method and results section highlighting the meaning of the different terms of jargon, parameters and metrics in a more plain language would make for smoother reading for an audience interested in this new way of monitoring coral physiology but less versed in computer sciences than the authors are. After all, the authors felt the need to define terms like semantic segmentation and fine-tuning so, they could expend on this didactic endeavor.

The discussion is short and makes minimal statements on the ecological implications of the potential application of this method in further ecological studies. This is not an issue but as a biologist who would potentially want to use this method to investigate corals activity in relation to their environment, I wonder how much work will be needed to re-create this method? Could you expend on how accessible this method would be?

Besides, this method seems heavily tailored to this specific case study given the method complexity, could all the steps be transferred to other cases without additional investigation and tuning?

Overall, opening-up to a less specialized audience by rephrasing or extending some passafes would help the diffusion of this method and better advocate its potential to the community that would use these results. I have made some suggestions towards better clarity of the text as well as some very minor typos in the text that should be corrected before publication.

14: on those

14-16: awkward phrasing

36: “data from both the cameras and other sensors provide a valuable… “

38 approaches in data

38: 5 data in 1 sentence. “the heterogeneity in the in the data calls for special approach…”

61: “the recorded time series spans less than a year”

67: break sentence in 2.

78: the monitoring of a larger area

81: method to extract

85: the question whether xxxx or not

105: If you are mentioning it, Can you succinctly define single features vectors? Their time lag?

110: target output = response variable that you are trying to predict? This is the pivotal paragraph where short clear sentences would make sure readers get the objectives.

186: not sure how the numbering works here as there are 13 lines between 186 and 187… Anyway, in that paragraph, it is stated 100 images were selected. Is that a random pick within all the available images and a further random split? Or was there an attempt to homogenize the representativity of the different states of the colonies in each one of the 3 sets?

Also, was it another 100 images selected for each intervals?

201: is there a reason for these specific numbers? – also, do you have an estimate of how much processing time was saved?

218: numbering stops again.

I feel it would help some of the audience unfamiliar with mathematical annotations to lay down the meaning of the equations in plain text to ensure the pivotal principles used in the study are clear.

Where does the 20% come from?

235: That is not an easy sentence to read.

250: is it all polyps retracted or there is no D patch where more than 20% of the polyps are extended?

288: to compute??

290: standardized to zero mean is zero and unit variance??? Does that need a rephrase?

295: odd start of a paragraph. Can you say what the sliding window will be used for?

482: rephrase sentence

488: periods of inactivity?

520: maybe but what is too much? Can you give some concrete element of what that entails?

520: rephrase sentence. e.g.: “It also indicates that transferring this approach to other CWC species has good chances of success”.

532: periods of camera failures

6. PLOS authors have the option to publish the peer review history of their article (what does this mean?). If published, this will include your full peer review and any attached files.

Reviewer #1: No

Reviewer #2: No

---

## [Author Response · Author response to Decision Letter 0]

20 Jan 2023

Please see the attached Response to Reviewers document.

---

## [Editor Report · Decision Letter 1]

22 Feb 2023

A data science approach for multi-sensor marine observatory data monitoring cold water corals (Paragorgia arborea) in two campaigns

PONE-D-22-23889R1

Dear Dr. Nattkemper,

We’re pleased to inform you that your manuscript has been judged scientifically suitable for publication and will be formally accepted for publication once it meets all outstanding technical requirements.

Kind regards,

Andrew Davies

Academic Editor

PLOS ONE
---

## [Editor Report · Acceptance letter]

28 Feb 2023

PONE-D-22-23889R1 

A data science approach for multi-sensor marine observatory data monitoring cold water corals (*Paragorgia arborea*) in two campaigns 

Dear Dr. Nattkemper:

I'm pleased to inform you that your manuscript has been deemed suitable for publication in PLOS ONE. Congratulations! Your manuscript is now with our production department. 

Kind regards, 

on behalf of

Dr Andrew Davies 

Academic Editor

PLOS ONE